# Conjugation with Phospholipids as a Modification Increasing Anticancer Activity of Phenolic Acids in Metastatic Melanoma—In Vitro and In Silico Studies

**DOI:** 10.3390/ijms22168397

**Published:** 2021-08-05

**Authors:** Anna Palko-Łabuz, Anna Gliszczyńska, Magdalena Skonieczna, Andrzej Poła, Olga Wesołowska, Kamila Środa-Pomianek

**Affiliations:** 1Department of Biophysics and Neuroscience, Wroclaw Medical University, ul. Chalubinskiego 3a, 50-368 Wroclaw, Poland; anna.palko-labuz@umed.wroc.pl (A.P.-Ł.); andrzej.pola@umed.wroc.pl (A.P.); kamila.sroda-pomianek@umed.wroc.pl (K.Ś.-P.); 2Department of Chemistry, Wrocław University of Environmental and Life Sciences, Norwida 25, 50-375 Wrocław, Poland; anna.gliszczynska@wp.pl; 3Department of Systems Biology and Engineering, The Silesian University of Technology, ul. Akademicka 16, 44-100 Gliwice, Poland; magdalena.skonieczna@polsl.pl; 4Biotechnology Centre, Silesian University of Technology, ul. Krzywoustego 8, 44-100 Gliwice, Poland

**Keywords:** metastatic melanoma, phenolic acids, phosphatidylcholine conjugates, anticancer activity, molecular modelling

## Abstract

Phenolic acids possess many beneficial biological activities, including antioxidant and anti-inflammatory properties. Unfortunately, their low bioavailability restricts their potential medical uses, as it limits the concentration of phenolic acids achievable in the organism. The conjugation with phospholipids constitutes one of the most effective strategies to enhance compounds bioavailability in biological systems. In the present study, the conjugates of anisic (ANISA) and veratric acid (VA) with phosphatidylcholine (PC) were investigated. Since both ANISA and VA are inhibitors of tyrosinase, a melanocyte enzyme, the expression of which increases during tumorigenesis, anticancer potential of the conjugates was tested in several metastatic melanoma cell lines. The conjugates proved to be antiproliferative, apoptosis-inducing and cell-cycle-affecting agents, selective for cancerous cells and not affecting normal fibroblasts. The conjugates substituted by ANISA and VA, respectively, at both the *sn*-1 and *sn*-2 positions of PC, appeared the most promising, since they were effective against the vast majority of metastatic melanoma cell lines. Additionally, the conjugation of phenolic acids to PC increased their antioxidant activity. Molecular modeling was employed for the first time to estimate the features of the investigated conjugates relevant to their anticancer properties and membrane permeation. Again, the conjugates substituted by phenolic acid at both the *sn*-1 and *sn*-2 positions of PC seemed to be presumably most bioavailable.

## 1. Introduction

Phenolic acids constitute an important class of plant secondary metabolites. They are ingested by humans with food, contributing to its nutritional properties, as well as taste and smell qualities. The consumption of high amounts of plant phenolics is believed to bring numerous health benefits. Especially, the antioxidant properties of phenolics and their ability to neutralize reactive oxygen species (ROS) contribute to their beneficial effects [1]. In the present study, the derivatives of anisic acid (4-methoxybenzoic acid; ANISA) and veratric acid (3,4-dimethoxybenzoic acid; VA) were investigated. VA is known from its anti-inflammatory activity and was demonstrated to reduce both the expression of inducible NO synthase [2] and the production of interleukins by lipopolysaccharide (LPS)-stimulated cells [3]. Additionally, VA was reported to protect human keratinocytes against UVB-induced damage [4], while ANISA was able to reduce the toxicity of various chemicals to hepatocytes [5]. ANISA was also reported to be the inhibitor of tyrosinase [6,7]. The same inhibitory activity was also demonstrated for the series of VA derivatives [8]. Expression of this enzyme is detected mainly in melanocytes and, consequently, in melanoma. Additionally, tyrosinase expression is known to increase during tumorigenesis [9]. For this reason, the use of strategies that target tyrosinase activity could allow for selective treatment of melanoma.

Even though, in comparison with other plant phenolics (e.g., flavonoids), free phenolic acids are relatively well water soluble [10], the low bioavailability limits their practical application as health-promoting substances in humans. The concentrations reached by phenolic acids in the organism highly depend on the level of their release from food matrix by the gut microbiota [11,12]. The fast metabolism of free phenolic acids and their rapid elimination in urine and bile after the ingestion additionally limits their bioavailability [12]. Therefore, chemical modification of phenolic acids, especially increasing their lipophilicity, could help in achieving the concentration of the active substances in the body to the levels required for their biological activity. Jóźwiak et al. (2020) indicated the possibility to increase bioavailability and anticancer activity of heterocyclic compounds via their conjugation with fatty acids [13].

The conjugation of the substance of interest with phospholipids constitutes one of the most effective strategies to enhance its bioavailability in biological systems [14]. This technique is commonly used to improve bioavailability and decrease systemic toxicity of anticancer drugs. Phospholipid conjugates of doxorubicin have been demonstrated to have improved absorption, better pharmacodynamic properties and increased selectivity between normal and cancerous cells [15,16]. In addition, the conjugation of tamoxifen to bile acid phospholipid yielded the derivative characterized by enhanced gut absorption, blood circulation and selective accumulation in tumor site that led to preferential anticancer activity in comparison with the parent drug [17]. Flavonoid-conjugates that possessed promising anticancer properties detected in in vitro studies were obtained by a covalent linkage of a flavonoid molecule to alkylphospholipid [18] and fatty acids [19]. Gliszczynska et al. [20,21] synthesized conjugates of isoprenoids with phosphatidylcholine (PC), a major component of eukaryotic cell membranes, and demonstrated the promising antiproliferative activities of the conjugates towards various types of cancer cells. Encouraged by the results, the authors produced conjugates in which cinnamic [22], anisic and veratric acids [23] substituted at position *sn*-1 and/or *sn*-2 of PC, also obtaining compounds of potent anticancer activity. On the other hand, it was proven that lysophosphatidylcholine bearing ANISA at the *sn*-1 position exhibited antidiabetic activity [24].

In the present study, the anticancer activity of the conjugates of anisic and veratric acid with PC were studied in several metastatic melanoma cells and compared with the activity in normal fibroblasts. The studied conjugates proved to be antiproliferative, apoptosis-inducing and cell-cycle-affecting agents selective for cancerous cells. The conjugates substituted by ANISA and VA, respectively, at both the *sn*-1 and *sn*-2 positions of PC, turned to be especially promising since they were effective against the vast majority of the studied metastatic melanoma cell lines. The conjugation of ANISA and VA to PC also resulted in an increase in anti-oxidative potency of the compounds. Molecular modeling was employed for the first time to estimate the features of the investigated conjugates relevant to their anticancer properties and membrane permeation. Again, the conjugates substituted by phenolic acid at both the *sn*-1 and *sn*-2 positions of PC seemed to be potentially the most bioavailable.

## 2. Results

### 2.1. Cell Growth Inhibition

The influence of pure ANISA (1a), VA (1b) and their conjugates with PC (3a, 5a, 8a, 3b, 5b and 8b) on the growth of melanoma (Me45, 451-Lu and 1205-Lu) and normal fibroblasts (NHDF) was studied. The compounds were tested in the concentration range from 50 to 400 μM and the time of incubation of the studied compound was 72 h. Free acids (1a and 1b) were identified as non-toxic agents, in this range of concentration. It was not possible to determine their IC_50_ values in any of the studied cell lines (Figure 1, Table 1). Conjugation of 1a or 1b with PC increased the growth inhibition potency of the studied phenolic acids. All conjugates exerted a more significant effect on cell growth, compared to pure acids (*p* < 0.05). Derivatives containing ANISA or VA at both the *sn*-1 and *sn*-2 positions of PC (3a and 3b) were found to be the most active compounds in 451-Lu and Me45 cells (Figure 1A). In contrast, these conjugates exhibited the lowest growth inhibitory potency in 1205-Lu and NHDF cells. In these lines, the highest antiproliferative effect was observed in the presence of derivatives containing palmitic acid at the *sn*-1 position and phenolic acid at the *sn*-2 position (8a and 8b) (Figure 1). It is worth noticing that of all studied conjugates exhibited significantly higher cytotoxic activity towards malignant melanoma than towards normal cells (Table 1, Appendix A).

Figure 1C also presents the representative images of melanoma cells (Me45 and 1205-Lu) cultured in the presence of 100 µM of non-active compound 1a and the most active 3a and 8a. The images conform to the results of MTT assay. The influence of free acid on the cells was negligible, while ANISA incorporated into PC structure altered both the confluence and cellular morphology. The presence of rounded and shrunken cells in 3a- and 8a-treated samples suggests that the cells might undergo apoptosis.

### 2.2. Reactive Oxygen Species Formation

Since phenolic acids are known for their antioxidant properties, the influence of free ANISA, VA and their conjugates with PC on intracellular reactive oxygen species formation was investigated in metastatic melanoma cells. The analysis of 2′,7′-dichlorofluorescein (DCF) fluorescence showed that free acids exhibited antioxidant activity in 451-Lu (Figure 2A) and 1205-Lu (Figure 2C) cells, whereas the effects observed in Me45 were negligible (Figure 2B). In normal fibroblasts, antioxidant activity of the studied conjugates was low or negligible (Appendix A). The results obtained for ANISA (1a) and VA (1b) were compared to the results obtained for their conjugates with PC. The experiments revealed that the antioxidant potency increased for phenolic acids incorporated into PC structure. In the case of ANISA conjugates, derivative 3a was identified as the strongest antioxidant in all studied cell lines, compared to pure acid (Figure 2A–C). Other conjugates enhanced antioxidant properties of ANISA only in 1205-Lu cells (Figure 2C). Free VA also slightly reduced ROS formation. Moreover, the conjugation of VA with PC strongly increased its antioxidant effects. Similarly to ANISA derivatives, the conjugates containing VA at both positions *sn*-1 and *sn*-2 of PC exhibited the highest potency to reduce the formation of ROS, compared to free VA (Figure 2A–C).

### 2.3. Cell Cycle Progression

To complete the study on the influence of phenolic acids and their conjugates on viability of melanoma cells, the analysis of cell cycle progression in their presence was performed. The influence of free ANISA (1a), VA (1b) and their conjugates with PC (3a, 5a, 8a, 3b, 5b and 8b) on cell cycle progression was studied after 72 h of incubation with the compound. Flow cytometry analysis revealed that all studied conjugates, but not free acids, increased the fraction of 1205-Lu cells in the sub-G1 phase, compared to the untreated control (Figure 3A,B). The sub-G1 phase represents death cells in the population, suggesting necrosis or/and late stages of apoptosis. Similar effects were observed in 451-Lu, but only in the presence of compounds 3a, 3b and 5b (Figure 4A,B). In contrast to 451-Lu and 1205-Lu cell lines, compound 3b increased the fraction of Me45 cells in the sub-G1 as well as G2/M phases, indicating not only the pro-apoptotic but also cytostatic potential of the compound (Figure 5A,B). The G2/M phase represents mitotically blocked cells with damaged DNA, lasting from the G1 and S phases or generated in G2. Interestingly, derivative 3a did not affect the Me45 cells cycle distribution, emphasizing the importance of an additional methoxy group in the structure of the molecule for the biological activity of compound in this cell line. Increased fraction of Me45 cells in G2/M phase was observed also in the presence of the 5b and 8b compounds, suggesting cell cycle arrest (Figure 5A,B). In contrast to melanoma cells, cell cycle of normal fibroblasts was not affected by any of the studied conjugates (Appendix A).

### 2.4. Apoptosis Induction

The ability of the studied compounds to induce apoptosis was also studied. 451-Lu, Me45, 1205-Lu and NHDF cells were incubated with phenolic acids and their conjugates for 72 h. Free acids (1a and 1b) had no ability to induce apoptosis in all studied melanoma cells. They also did not increase the number of necrotic cells. However, derivatives 3a and 3b, which have phenolic acids incorporated into the positions *sn*-1 as well as *sn*-2 of PC, showed significant potency to induce apoptosis in 451-Lu, Me45 and 1205-Lu cells (Figure 6, Figure 7 and Figure 8). A higher fraction of cells in the late stage of apoptosis, compared to the untreated control, was recorded, whereas the change in the amount of the cells in early apoptotic phase was negligible. In the case of 451-Lu and 1205-Lu, double-substituted conjugates (3a and 3b) also caused a significant increase in the fraction of necrotic cells (Figure 6A,B and Figure 7A,B). The activity of all other derivatives as apoptosis and/or necrosis inducers was noticeable only in 1205-Lu cells (Figure 7A). However, no clear relationship between the chemical structure of the derivatives and their pro-apoptotic activity was observed. The conjugate containing ANISA at the *sn*-2 position (5a) slightly increased the percentage of 451-Lu (Figure 6A) and Me45 cells (Figure 8A) in late apoptosis (*p* < 0.05). Its corresponding derivative, possessing VA in the structure (5b), induced a similar effect only in 451-Lu cells (Figure 6A). Moreover, significantly higher fractions of Me45 cells in the early and late stages of apoptosis were observed in the presence of this compound, compared to the control (Figure 8A,B). On the other hand, none of the studied conjugates was able to induce apoptosis in NHDF cells (Appendix A), which pointed to their selective activity towards melanoma cells.

In the next step, pilot experiments on the pro-apoptotic activity of phenolic acid-PC conjugates in combination with ionizing radiation (IR) were performed. Since the most pronounced pro-apoptotic effects of studied compounds were observed in 1205-Lu cells, this cell line was selected as a model for the experiment. The time of incubation with a given compound was reduced to 48 h. X-rays at the dose of 2 Gy were applied at the 24th hour of incubation.

Pro-apoptotic effects of the conjugates after 48 h of incubation were weaker than after 72 h. The highest fractions of cells in the early stage of apoptosis, ~17% and ~22%, were noted in the presence of 3a and 3b derivatives, respectively (Figure 9). The amounts of cells undergoing the later steps of programmed cell death or necrosis were negligible (data not shown). Irradiation of 1205-Lu cells resulted in the increase in early apoptotic but not late apoptotic and necrotic cell fractions. Such an increase was recorded both in the untreated control and in conjugate-treated samples. However, the pre-treatment of cells with the derivatives of phenolic acids before irradiation caused more significant changes in the number of apoptotic cells. In the presence of most compounds, the percentage of apoptotic 1205-Lu cells was over 70%. The most effective combinations included ionizing radiation and the conjugates containing phenolic acid at the *sn*-1 position (8a and 8b) (Figure 9).

### 2.5. Molecular Calculations

For a better understanding of the molecular features responsible for superior anticancer activity of the conjugates, compared to free phenolic acids, molecular modelling was applied. The results are presented in Table 2. The energy difference ΔE between the HOMO (highest occupied molecular orbital) and LUMO (lowest unoccupied molecular orbital) energy levels, called energy gap, indirectly informs about compound reactivity. Generally, the energy gaps for free phenolic acids and PC-conjugates were comparable. In the case of ANISA derivatives, conjugation slightly increased ΔE values, whereas, in the case of VA free acid, it had a greater energy gap than its conjugates (with exception of 8b). The conjugation of phenolic acids to PC yielded derivatives of reduced electronegativity, which is related to electron donating properties of the molecule. In addition, the ability of the conjugates to accept electrons, as measured by electrophilicity, was lower, in comparison to free phenolic acids. On the other hand, dipole moments of all the conjugates were much higher than those of 1a and 1b, which pointed to increased polarity of the derivatives. The octanol:water partition coefficient is the measure describing the preference of a molecule to dissolve in water or in organic solvent phase. As can be noticed, 1a and 1b showed no preference, while the conjugates containing the palmitoyl chain were more lipohilic than the parent compounds. The derivatives 3a and 3b exhibited high hydrophilicity. This notion was supported by the obtained values of solubility, which clearly pointed to the highest water solubility of 3a and 3b, followed by ANISA and VA, while derivatives 5a, 5b, 8a and 8b were weakly water soluble. Moreover, compounds 3a and 3b had the highest topological polar surface area of all the investigated conjugates.

Further analysis prompted us to look at the spatial orientation of HOMO and LUMO orbitals in the molecules of phenolic acids and their conjugates. As can be seen in Figure 10, in the ANISA molecule, HOMO orbitals were concentrated mainly at its aromatic ring, while LUMO orbitals were found around the chemical bonds directly attached to the ring. The process of conjugation of phenolic acid to PC did not affect much the distribution of HOMO and LUMO orbitals. As shown in the example of the 3a conjugate, HOMO and LUMO orbitals remained at the phenolic acid fragments of the conjugate. The examination of the electrostatic potential map of 1a (Figure 10E) revealed that the highest negative potential was observed near O atoms from an acid group, while H atoms of the same group, as well as the CH_3_ group at the side chain, remained the centers of positive potential. The conjugation of ANISA to PC dramatically changed the appearance of the electrostatic potential map (Figure 10F). The most negative region of the molecule was now centered near the P atom and the most positive region was close to the N atom. The most peripheral CH_3_ groups also possessed slightly positive potential.

## 3. Discussion

The results of the performed experiments brought evidence of anticancer activity of the conjugates of phenolic acids with PC. It was evident that the conjugates possessed superior cell growth inhibitory activity to free compounds ANISA (1a) and VA (1b). The anticancer potency of the studied conjugates depended on the cell line, since different compounds were identified to be the most effective in various melanoma cell lines. Moreover, IC_50_ values of the conjugates recorded in malignant cells were at least 1.5–2 lower than the IC_50_ values obtained in normal fibroblasts. The selectivity between cancer and non-cancer cells was therefore observed for the studied derivatives of phenolic acids. The cytotoxicity of the same conjugates has already been studied in human leukemia, breast, lung, liver and colon cancer cells [23]. As a rule, the IC_50_ values of the conjugates were lower than the IC_50_ of pure acids. Interestingly, compound 3b was also able to partially overcome doxorubicin-resistance in colon cancer cells. Similarly, superior anticancer activity of phospholipid conjugates, compared to free compounds, was observed for cinnamic and 3-methoxycinnamic acids [22], as well as isoprenoids [21]. In addition, conjugation of quercetin with alkylphospholipids increased the cytotoxic activity of the obtained derivatives towards human liver, pancreatic and ovarian cancer cells [18]. On the other hand, tamoxifen conjugated to lithocholic-acid-derived phospholipid was less cytotoxic to breast cancer cells, compared to pure drug, but its selectivity for cancer cells over normal cells was increased [17].

Since beneficial effects of phenolic acids are believed to be due to their antioxidant activities [1], the influence of the studied conjugates on intracellular ROS level in melanoma cells was studied. Both free ANISA (1a) and VA (1b) declined ROS formation; however, their potency was strongly dependent of the cell line used, with 1205-Lu being the most sensitive and Me45 being the least. The conjugation of phenolic acids to PC increased their antioxidant activity, that was more pronounced in the case of VA-derivatives than in that of derivatives of ANISA. For both phenolic acids, the conjugated substituted at both the *sn*-1 and *sn*-2 positions (3a and 3b) exhibited the highest ability to reduce ROS formation. In one of the previous studies, it was shown that the conjugation of grapefruit extract, naringin and neohesperidin dihydrochalcone with fatty acids yielded compounds with comparable antioxidant activity to that of free flavonoids [19]. Only the derivatives containing omega-3 polyunsaturated fatty acids possessed higher activity than the respective parent compounds.

The analysis of the effect of the studied compounds on the cell cycle of melanoma cells revealed that the unconjugated phenolic acids did not change the distribution of melanoma cells between different phases of the cell cycle. Similarly, none of the studied compounds affected the cell cycle of normal fibroblasts. The conjugates increased the fraction of melanoma cells in the sub-G1 phase, compared to the untreated control, that suggested the appearance of apoptotic and/or necrotic cells in the population. This was visible for all conjugates in 1205-Lu cells, for 3a, 3b and 5b in 451-Lu cells and only for compounds 3a and 3b in Me45 cells. Additionally, the derivative 3b induced the arrest in the G2/M phase, pointing to the cytostatic potential of the compound. The increase in the number of breast cancer cells in the sub-G1 phase of the cell cycle was also observed for tamoxifen and its phospholipid-conjugate [17]. The conjugate of cinnamic acid with PC studied in leukemia cells induced cell cycle arrest in the G2/M phase [22], while the PC-conjugate of ANISA (other than studied here) arrested cell cycle in the G0/G1 phase [23].

Since the studied conjugates were suspected to induce apoptosis, the relevant studies were performed. The studied compounds (at 100 µM) induced virtually no apoptosis in normal fibroblasts. The experiments revealed that all the conjugates (but not free phenolic acids) increased the number of apoptotic and necrotic cells in the 1205-Lu cell line. In 451-Lu cells only compounds 3a and 3b exhibited apoptosis-inducing potency, while, in Me45 cells, 3a, 3b, 5a and 5b were identified as apoptosis inducers. Additionally, all the conjugates significantly enhanced apoptosis induced by the application of ionizing radiation in 1205-Lu cells. The previously studied conjugate of cinnamic acid with PC was observed to reduce mitochondrial potential, but no apoptosis induction was revealed by means of an Annexin-V assay [22]. Since the derivative caused the increase in the number of cells in the G2/M phase with no apoptosis-inducing potency, the authors concluded that the derivative acted as a cytostatic rather than cytotoxic agent in leukemia cells. Two conjugates of quercetin with alkylphospholipids were demonstrated to increase the size of apoptotic cell population in liver cancer cells, to down-regulate the expression of anti-apoptotic protein Bcl-2 and also facilitate the cleavage of poly (ADP-ribose) polymerase (PARP) by caspases [18].

Molecular modelling was performed in hope to distinguish the structural features that might be responsible for superior anticancer properties of the studied conjugates, compared to free phenolic acids. Generally, all the conjugates possessed lower electronegativity, lower electrophilicity but higher polarity than their respective phenolic acids. The most distinctive features were found in compounds 3a and 3b. These conjugates turned out to be effective anticancer agents in the vast majority of the studied melanoma cell lines. Both were characterized by high hydrophilicity and possessed the largest topological polar surface area. These inevitably affected their solubility and permeation through cellular membranes. The molecular polar surface area has been shown to correlate with passive transport through membranes and for this reason it is used to predict intestinal absorption [25,26] and blood-brain barrier penetration [27]. Additionally, the conjugation of phenolic acids with PC strongly affected electrostatic potential distribution on the molecule surface with most positive and most negative regions present near N and P atoms of PC in the conjugates, respectively.

## 4. Materials and Methods

### 4.1. Conjugates

Free anisic acid (1a) and veratric acid (1b) were purchased from Sigma-Aldrich (Poznan, Poland). Phosphatidylcholines with O-methylated derivatives of phenolcarboxylic acids, 1,2-dianisoyl-*sn*-glycero-3-phosphocholine (3a), 1,2-diveratroyl-*sn*-glycero-3-phosphocholine (3b), 1-palmitoyl-2-anisoyl-*sn*-glycero-3-phosphocholine (5b), 1-anisoyl-2-palmitoyl-*sn*-glycero-3-phosphocholine (8a) and 1-veratroyl-2-palmitoyl-*sn*-glycero-3-phosphocholine (8b), were synthesized in the Department of Chemistry, Wroclaw University of Environmental and Life Sciences, according to the procedure described previously [23]. Stock solutions of all compounds were prepared in the mixture ethanol: DMSO (*v/v*, 1:1; both purchased from Sigma-Aldrich, Poznan, Poland) and were then diluted in cell culture media. Chemical structures of the studied compounds are presented in Figure 11.

### 4.2. Cell Cultures

Three malignant melanoma cell lines (Me45, 1205-Lu and 451-Lu; obtained from the collection of the Centre of Oncology, Gliwice, Poland), neonatal human dermal fibroblast (NHDF; Lonza, Basel, Switzerland) and human colorectal adenocarcinoma cells (Caco-2; ATCC, Manassas, VA, USA) were used in the studies. The human malignant melanoma (Me45) cell line (derived from a lymph node metastasis of skin melanoma) was established in 1997 at the Radiobiology Department of the Centre of Oncology in Gliwice. 1205-Lu and 451-Lu cell lines were derived from human melanoma metastases to lungs of immunodeficient mice [28]. The cells were cultivated in the medium DMEM-F12 (PAA, Warsaw, Poland) supplemented with 10% (for Me45, 1205-Lu and 451-Lu and NHDF) or 20% (Caco-2) of fetal bovine serum (FBS, Eurx, Gdansk, Poland). In addition, medium for Caco-2 cells contained Non-essential Amino Acid Solution (1×) (Sigma-Aldrich, Poznan, Poland). All cell lines were grown in the presence of 1% of antibiotics (10,000 μg/mL streptomycin and 10,000 units/mL penicillin; Sigma-Aldrich, Poznan, Poland), at 37 °C in humidified atmosphere with 5% CO_2_ under standard conditions. All experimental procedures were carried out in log-phase of cell growth. During some experiments melanoma cells were irradiated with X-rays at the dose of 2 Gy at the 24th hour of the incubation. Ionizing radiation was applied using a laboratory X-Cell 140 irradiator (Kubtec, Bridgeport, CT, USA).

### 4.3. Cell Viability Assay

Cells (Me45, 1205-Lu, 451-Lu and NHDF) were seeded on 96-well plates (20,000 cells/mL) 24 h before the experiment. One day later, fresh media containing tested compounds were added. The appropriate control, containing ethanol:DMSO (*v/v* 1:1) in fresh medium, was also prepared. The plates were incubated for a given time (72 h for Me45, 1205-Lu, 451-Lu and NHDF; 2 h for Caco-2) at 37 °C in humidified atmosphere with 5% CO_2_. After incubation, the medium was removed and cells were washed with physiological saline (PBS; PAA, Warsaw, Poland). In the next step, the cells were incubated with MTT (3-[4,5-dimethylthiazol-2-yl]-2,5-diphenyltetrazolium bromide, Sigma-Aldrich, Poznan, Poland) (solution of 0.5 mg/mL in RPMI 1640 without phenol red) for 2 h. Then, the MTT solution was removed and the formazan crystals were dissolved in the mixture isopropanol:HCl (*v/v* 1:0.04; both purchased from Sigma-Aldrich, Poznan, Poland). The absorbance of the obtained product was read at 570 nm using a microplate reader (Epoch; BioTek, Winooski, VT, USA). Survival rate was expressed as the percentage of cell survival and calculated from the ratio (A_570_ of treated cells/A_570_ of control cells) × 100%. The experiments were repeated three times.

### 4.4. Flow Cytometry Analysis

The studies of cell cycle, apoptosis and intracellular level of ROS were performed using the flow cytometry method. For the apoptosis assay, the cells were harvested and centrifuged (1500 rpm, 3 min). The apoptosis was measured using an Annexin-V apoptosis assay (BioLegend, San Diego, CA, USA). After supernatant removal, the cells were washed with PBS and centrifuged again. The pellets of cells were suspended in 50 μL of cold Annexin-V binding buffer and stained with FITC-labeled Annexin-V for 30 min at 37 °C in the dark. The Annexin-V binding buffer and propidium iodide (PI) solution (100 μg/mL; Sigma-Aldrich, Poznan, Poland) were then added and the samples were stored on ice till they were measured.

For cell cycle analysis, harvested cells were centrifuged (1500 rpm, 3 min). After supernatant removal, cells were fixed with a hypotonic buffer, containing PI at 100 µg/mL in PBS, 5 mg/l of citric acid, 1:9 Triton-X solution, 100 µg/mL of RNase in PBS (Sigma-Aldrich, Poznan, Poland). Probes were incubated for 20 min at room temperature in the dark, then stored on ice till they were measured.

The experiments on ROS content were performed after 72 h of incubation with the given compounds. The cells were harvested and centrifuged (1500 rpm, 3 min), washed with PBS and centrifuged again. The pellet was resuspended in culture medium and a cell-permeable non-fluorescent probe 2′,7′-dichlorofluorescin diacetate (DCFH-DA, Sigma-Aldrich, Poznan, Poland) at a final concentration of 30 µM was added. The samples were incubated in darkness for 30 min at 37 °C, then stored on ice till they were measured. DCFH-DA is hydrolyzed by cellular esterases into its polar form DCFH. Due to the intracellular ROS and other peroxides, DCFH undergoes oxidation and turns into highly fluorescent compond 2′,7′-dichlorofluorescein (DCF).

Cytometric analyses were performed immediately using an Aria III flow cytometer (Becton Dickinson, Franklin Lakes, NJ, USA) with FITC configuration (488 nm excitation; emission, LP mirror 503, BP filter 530/30) or with PE configuration (547 nm excitation; emission, 585 nm) and at least 10,000 cells were counted.

### 4.5. Molecular Modelling

Quantum calculations were performed using the SPARTAN’18 software (Wavefunction, Inc., Irvine, CA, USA). Geometry of the molecule of each compound in aqueous phase was optimized using the density functional theory (DFT) method with ωB97X-D exchange correlation potential (including empirical corrections for dispersive interactions) in connection with 6-311 + G ** basic set. The optimized geometry of the given molecule was confirmed to be the real minimum by frequency analysis (no imaginary frequencies). The solvent effect on geometry and value of quantum mechanical parameters was assessed using the conductor-like polarizable continuum model (CPCM) method. The values of absolute electronegativity (*χ*) and electrophilicity index (*ω*) were calculated on the basis of the calculated energy value for HOMO (*E_HOMO_*) and LUMO orbitals (*E_LUMO_*) using the following equations [29,30]:(1)$$\chi =-\frac{E_{LUMO}+E_{HOMO}}{2}$$
(2)$$\varpi =\frac{\chi }{E_{LUMO}-E_{HOMO}}$$

The physicochemical properties of the studied molecules were obtained using the OSIRIS property explorer [31]. Drug-relevant properties, including the octanol: water partition coefficient (cLogP), solubility (LogS), topological polar surface area and molecular weight, were estimated.

### 4.6. Statistical Analysis

The values of measured parameters are presented as the means ± standard deviation (SD) from three independent experiments. The statistical significance was determined by Student’s *t*-test (* *p* < 0.05 and ** *p* < 0.001) with the use of the Statistica 10 software (Tibco Software Inc., Palo Alto, CA, USA).

## 5. Conclusions

In conclusion, the conjugates of ANISA and VA with PC were demonstrated to be effective anticancer agents in metastatic melanoma cell lines and not to affect normal fibroblasts. The conjugates 3a and 3b, substituted by ANISA and VA, respectively, at both the *sn*-1 and *sn*-2 positions, appeared the most promising, since they were effective against the vast majority of the studied cancer cell lines. The conjugates induced apoptosis and affected cell cycle in metastatic melanoma cells. Additionally, the conjugation of phenolic acids to PC increased their antioxidant activity. The methods of molecular modelling used to point to the physicochemical properties of the studied compounds relevant to their anticancer properties and putative membrane permeation again distinguished conjugates 3a and 3b as presumably the most bioavailable.

## Figures and Tables

**Figure 1 ijms-22-08397-f001:**
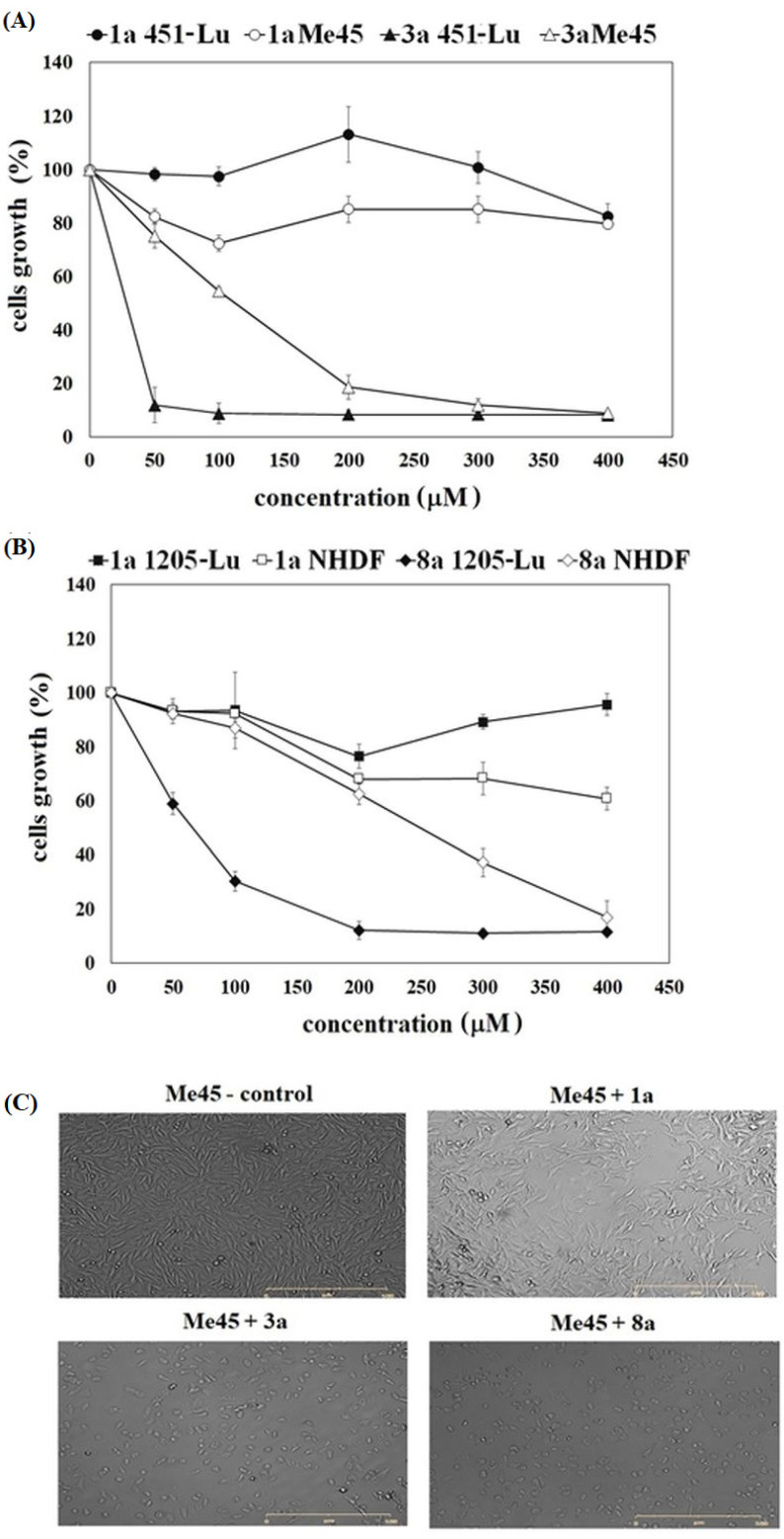
The influence of free phenolic acid (1a) and its conjugates on the studied cell lines. Representative profiles of growth inhibition exerted by free ANISA and the most active ANISA-conjugates in 451-Lu and Me45 (**A**), as well as in 1205-Lu and NHDF cells (**B**). Representative images of Me45 and 1205-Lu melanoma cells in the presence of ANISA and its derivatives at 100 µM (**C**).

**Figure 2 ijms-22-08397-f002:**
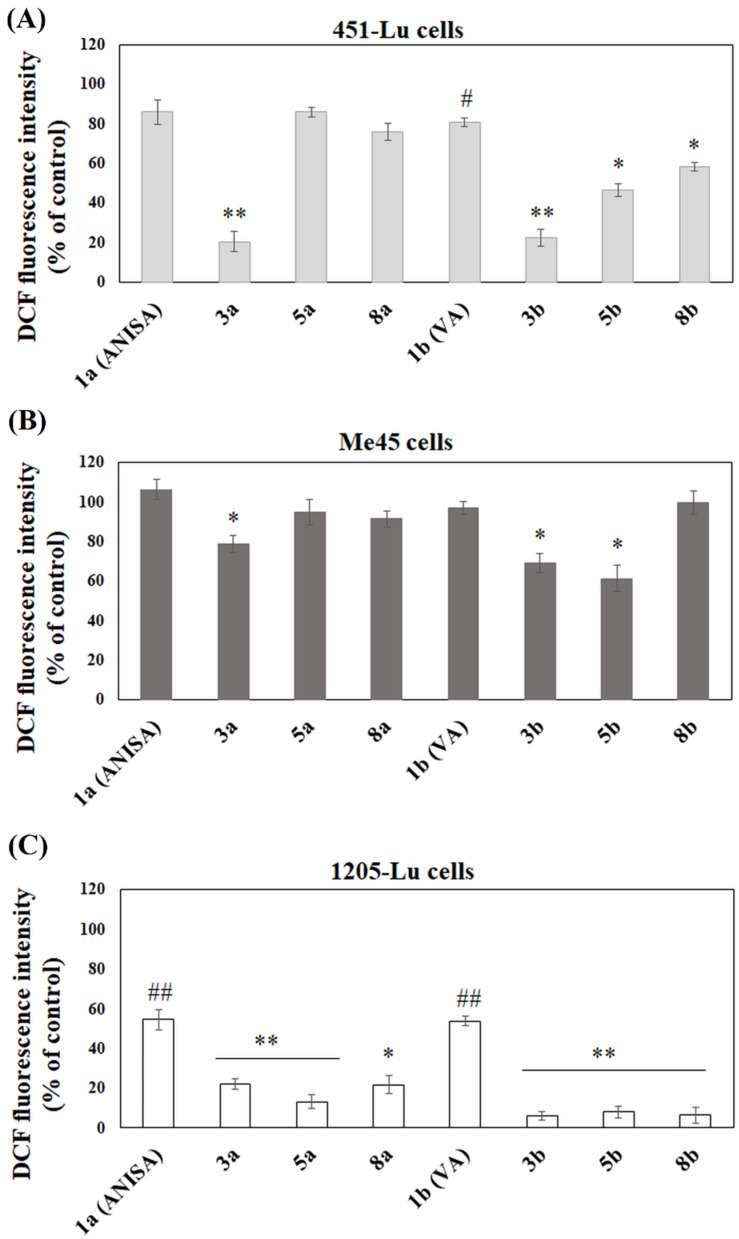
The influence of ANISA, VA and their conjugates on the reactive oxygen species formation in 451-Lu (**A**), Me45 (**B**) and 1205-Lu (**C**) melanoma cells. The results were presented as percentages of the control (no compound). In the case of the free acids ANISA and VA, statistical significance was compared to the samples containing no compounds (# *p* < 0.05; ## *p* < 0.001), whereas, in the case of PC-conjugates, to the samples containing ANISA or VA (* *p* < 0.05; ** *p* < 0.001).

**Figure 3 ijms-22-08397-f003:**
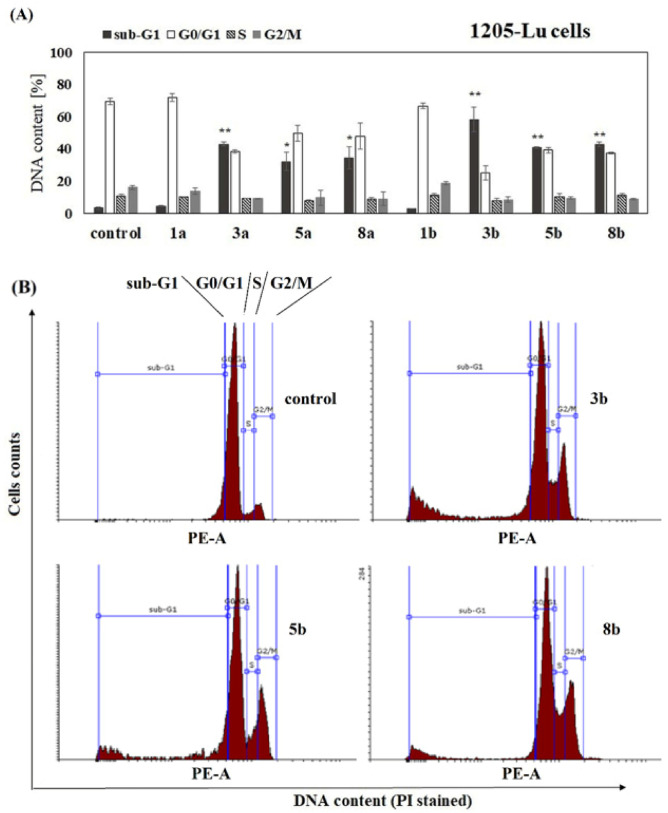
The distribution of 1205-Lu cells in major phases of the cycle. Cells were cultured for 72 h in the presence of the studied chemicals at a concentration of 100 µM. The statistical significance was determined (* *p* < 0.05 and ** *p* < 0.001) (**A**). Representative histograms for effects obtained in 1205-Lu cell line are shown (**B**).

**Figure 4 ijms-22-08397-f004:**
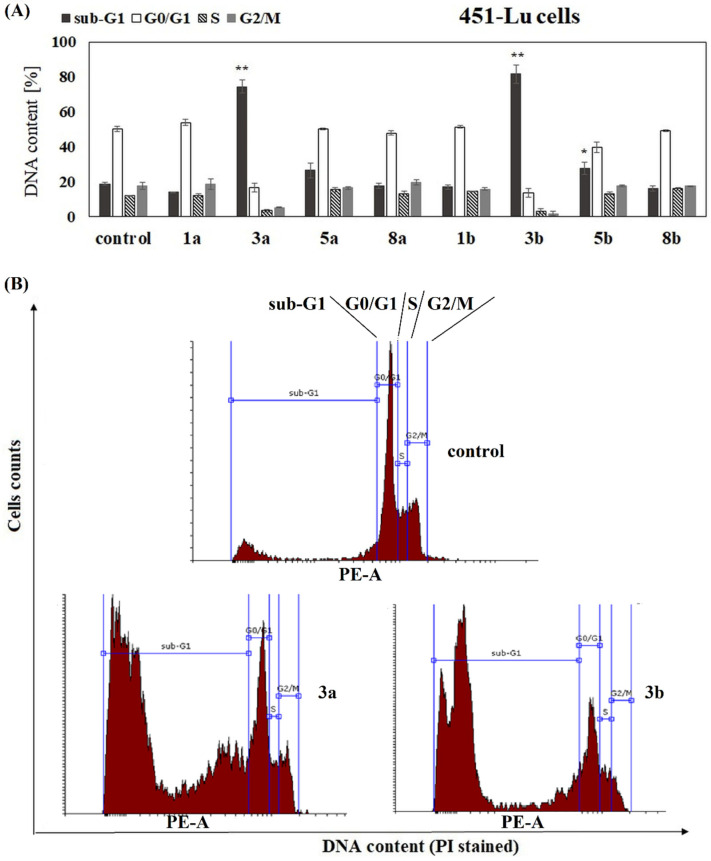
The distribution of 451-Lu cells in major phases of the cycle. Cells were cultured for 72 h in the presence of the studied chemicals at a concentration of 100 µM. The statistical significance was determined (* *p* < 0.05 and ** *p* < 0.001) (**A**). Representative histograms for effects obtained in 451-Lu cell line are shown (**B**).

**Figure 5 ijms-22-08397-f005:**
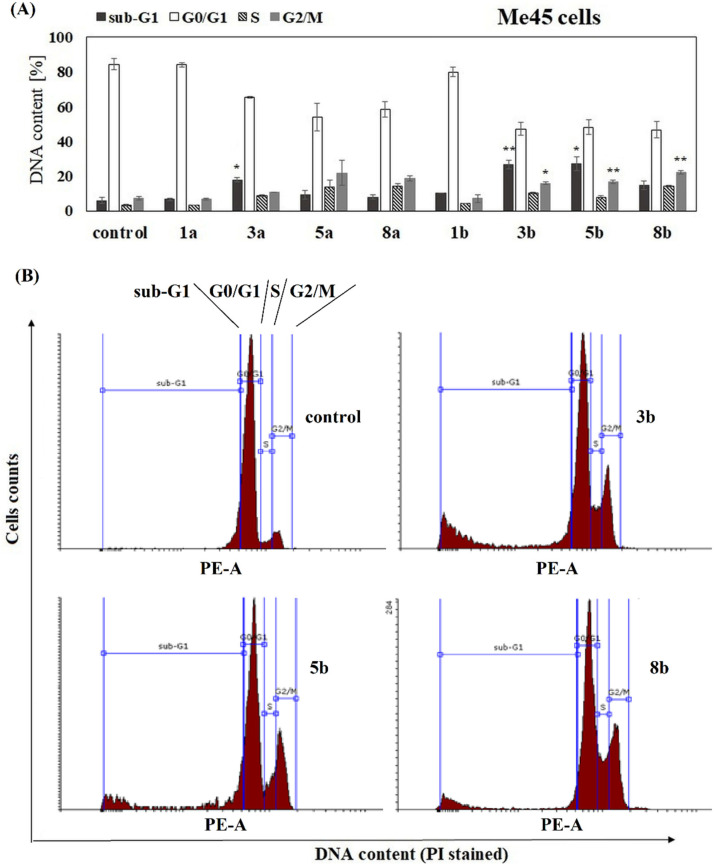
The distribution of Me45 cells in major phases. Cells were cultured for 72 h in the presence of the studied chemicals at a concentration of 100 µM. The statistical significance was determined (* *p* < 0.05 and ** *p* < 0.001) (**A**). Representative histograms for effects obtained in Me45 cell line are shown (**B**).

**Figure 6 ijms-22-08397-f006:**
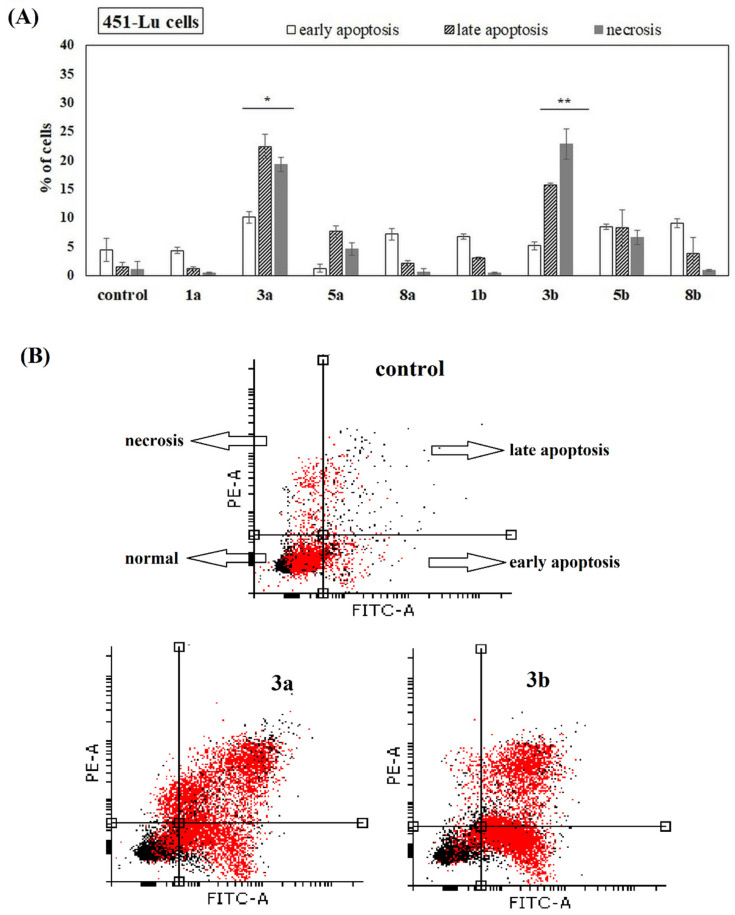
Induction of apoptosis in 451-Lu cells in the presence of the studied compounds. Cells were cultured for 72 h in the presence of the studied chemicals (at 100 µM). The percentage of necrotic, early and late apoptotic cells was shown. The statistical significance was determined (* *p* < 0.05 and ** *p* < 0.001) (**A**). Representative histograms of events defined after forward and side scatters as cells (red dots) and debris (black dots) are shown (**B**).

**Figure 7 ijms-22-08397-f007:**
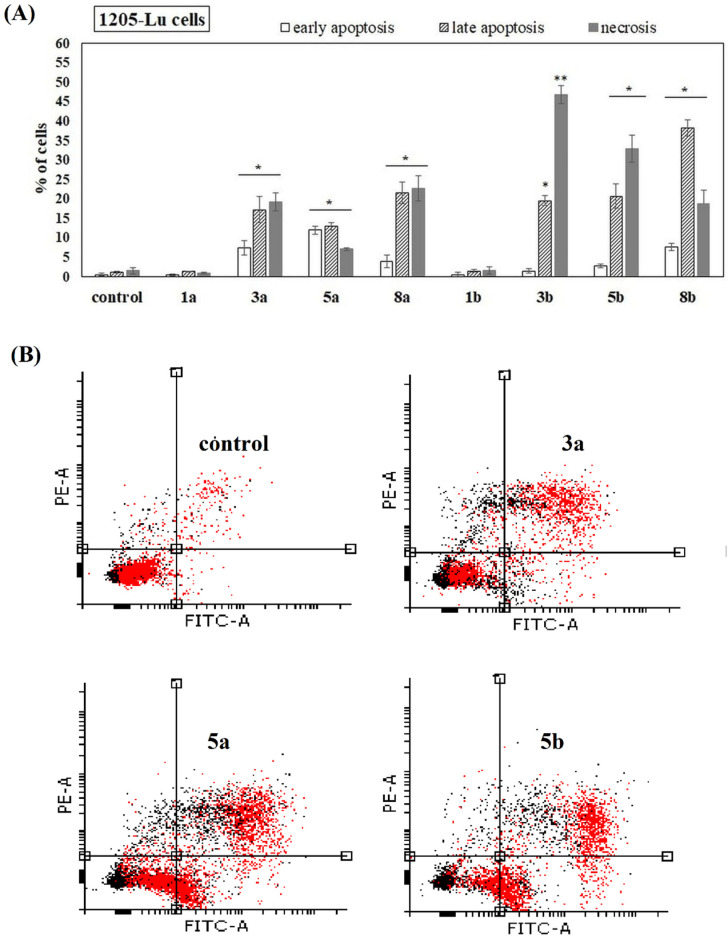
Induction of apoptosis in 1205-Lu cells in the presence of the studied compounds. Cells were cultured for 72 h in the presence of the studied chemicals (at 100 µM). The percentage of necrotic, early and late apoptotic cells was shown. The statistical significance was determined (* *p* < 0.05 and ** *p* < 0.001) (**A**). Representative histograms of events defined after forward and side scatters as cells (red dots) and debris (black dots) are shown (**B**).

**Figure 8 ijms-22-08397-f008:**
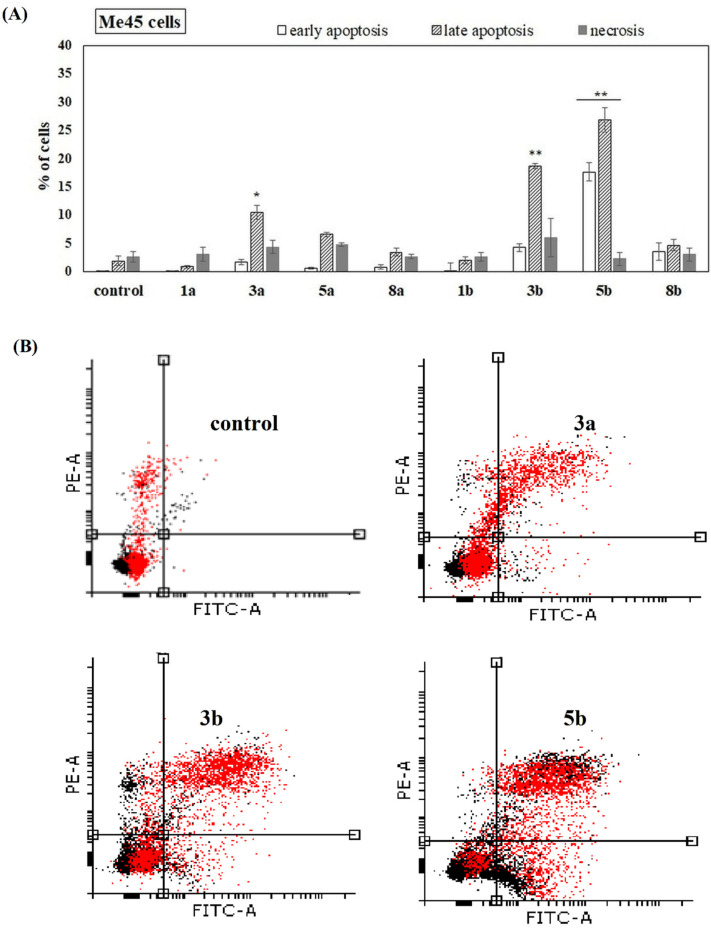
Induction of apoptosis in Me45 cells in the presence of the studied compounds. Cells were cultured for 72 h in the presence of the studied chemicals (at 100 µM). The percentage of necrotic, early and late apoptotic cells was shown. The statistical significance was determined (* *p* < 0.05 and ** *p* < 0.001) (**A**). Representative histograms of events defined after forward and side scatters as cells (red dots) and debris (black dots) are shown (**B**).

**Figure 9 ijms-22-08397-f009:**
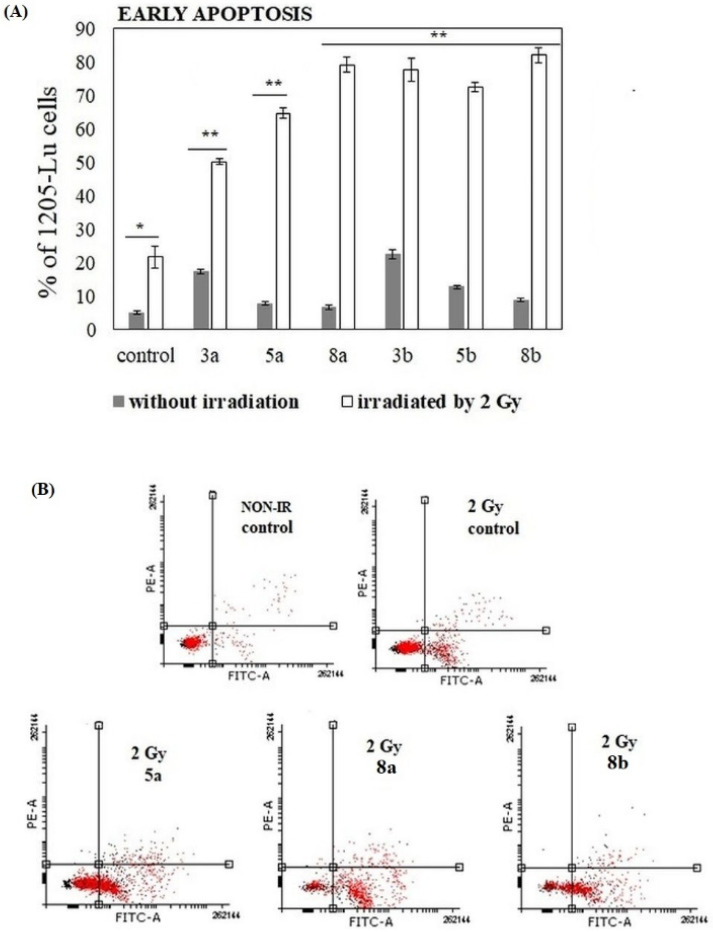
Pro-apoptotic effects of phenolic acid–PC conjugates in combination with ionizing in 1205-Lu cells. The percentages of early apoptotic cells pre-incubated with 100 µM of the compounds (grey bars) were compared to the cells that were also irradiated (2 Gy) (white bars). The statistical significance was determined (* *p* < 0.05 and ** *p* < 0.001) (**A**). Representative histograms of events defined after forward and side scatters as cells (red dots) and debris (black dots) are shown (**B**).

**Figure 10 ijms-22-08397-f010:**
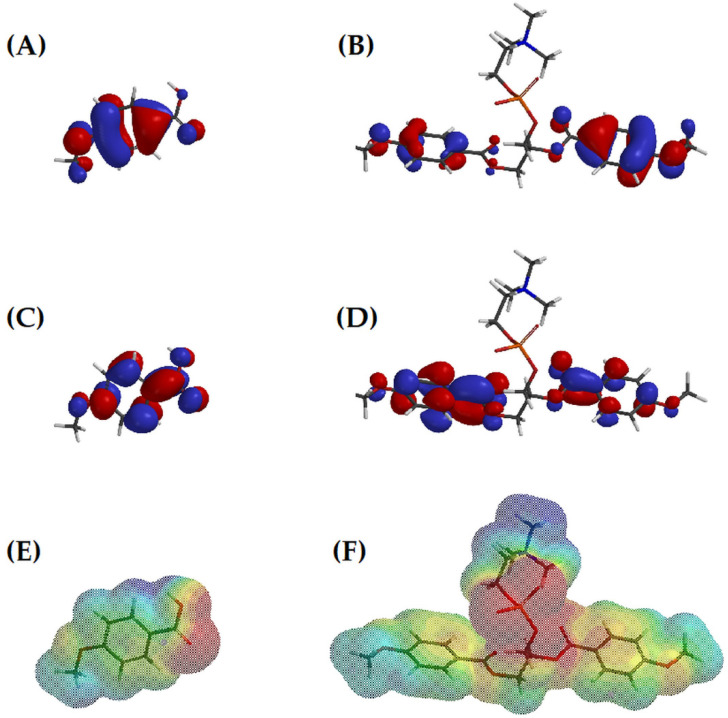
The spatial orientation of HOMO (**A**,**B**) and LUMO (**C**,**D**) orbitals and electrostatic potential map (**E**,**F**) of ANISA (1a, left column) and conjugate 3a (right column) in aqueous solution (the blue color represents the positive region and the red color represents the negative region).

**Figure 11 ijms-22-08397-f011:**
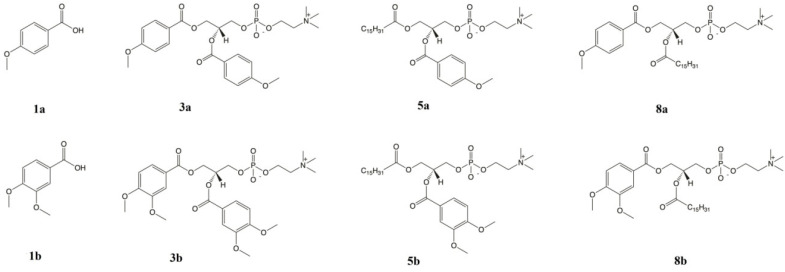
Studied conjugates of phospholipids and methoxy derivatives of benzoic acid: *p*-anisic acid (ANISA; **1a**), 1,2-dianisoyl-*sn*-glycero-3-phosphocholine (**3a**), 1-palmitoyl-2-anisoyl-*sn*-glycero-3-phosphocholine (**5a**), 1-anisoyl-2-palmitoyl-*sn*-glycero-3-phosphocholine (**8a**), veratric acid (VA; **1b**), 1,2-diveratroyl-*sn*-glycero-3-phosphocholine (**3b**), 1-palmitoyl-2-veratroyl-*sn*-glycero-3-phosphocholine (**5b**) and 1-veratroyl-2-palmitoyl-*sn*-glycero-3-phosphocholine (**8b**).

**Table 1 ijms-22-08397-t001:** The half maximal inhibitory concentrations (IC_50_) of studied compounds in metastatic melanoma cells (451-Lu, 1205-Lu and Me45) and normal skin fibroblasts NHDF.

Compound	Acyl Residue		Cell Line IC_50_ (μM)			
	*sn*-1	*sn*-2	451-Lu	1205-Lu	Me45	NHDF
1a (ANISA)	-	-	n.a.	n.a.	n.a.	n.a.
3a	ANISA	ANISA	47.5 ± 2.7	82.0 ± 10.5	111.20 ± 4.5	238.0 ± 17.8
5a	ANISA	PA ^2^	97.7 ± 8.0	71.3 ± 7.7	122.4 ± 10.0	249.0 ± 20.9
8a	PA	ANISA	54.1 ± 6.7	62.2 ± 7.1	130.9 ± 9.9	197.7 ± 20.8
1b (VA)	-	-	n.a.	n.a.	n.a.	n.a.
3b	VA	VA	48.3 ± 5.5	153.1 ± 9.5	84.4 ± 14.0	168.3 ± 11.3
5b	VA	PA	136.6 ± 7.7	102.0 ± 4.1	108.7 ± 7.6	184.8 ± 13.7
8b	PA	VA	82.2 ± 14.3	73.2 ± 9.3	126.1 ± 3.7	150.2 ± 17.9

n.a., not available; ANISA, anisic acid; VA, veratric acid; PA, palmitic acid.

**Table 2 ijms-22-08397-t002:** Molecular descriptors obtained for phenolic acids and their conjugates with PC.

Descriptor	1a	3a	5a	8a	1b	3b	5b	8b
E_HOMO_ (eV) ^1^	−6.70	−6.28	−6.31	−6.29	−6.59	−6.10	−6.11	−6.28
E_LUMO_ (eV) ^2^	−1.62	−1.02	−1.19	−1.17	−1.63	−1.29	−1.27	−1.20
ΔE (eV) ^3^	5.08	5.08	5.12	5.12	4.96	4.81	4.84	5.08
Electronegativity (eV)	−8.32	−7.48	−7.50	−7.46	−8.22	−7.39	−7.38	−7.48
Electrophilicity (eV)	13.63	11.01	10.99	10.87	13.62	11.35	11.25	11.01
Dipole moment (D)	7.41	16.39	20.06	16.50	7.70	20.31	16.59	15.85
cLog P ^4^	1.07	−4.02	1.43	1.43	1	−4.16	1.36	1.36
Solubility (LogS)	−1.65	−1.17	−4.03	−4.03	−1.66	−1.21	−4.05	−4.05
TPSA ^5^	46.53	139.46	130.23	130.23	55.76	157.92	139.46	139.46
MW ^6^	152.15	526.18	630.38	630.38	182.17	586.21	660.39	660.39

^1^ E_HOMO_, energy of the highest occupied molecular orbital; ^2^ E_LUMO_, energy of the lowest unoccupied molecular orbital; ^3^ ΔE = E_LUMO_ − E_HOMO_; ^4^ cLog P, octanol:water partition coefficient; ^5^ TPSA, topological polar surface area; ^6^ MW, molecular weight.

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
