# Peer review of "Conjugation with Phospholipids as a Modification Increasing Anticancer Activity of Phenolic Acids in Metastatic Melanoma—In Vitro and In Silico Studies"

_ijms, 2021, doi:10.3390/ijms22168397_

Round 1

Reviewer 1 Report

Conjugation with phospholipids as a modification increasing anticancer activity of phenolic acids in metastatic melanoma – in vitro and in silico studies

Comments

Introduction

I think that they should update the bibliography, which is a little outdated,

Line 41 -For example regarding the antioxidant properties [1] dates from 1999 and there is recent bibliography about this subject. I propose updated the bibliography

Line 75-76 “Recently Gliszczynska et al. [20,21]”- These articles are dated 2016-2017, I think that is not recently?

Results

Line 102 – “and 2a” – check if it is 2a or 1b ?

Line 111- “8a and 8b- figure 1B “ -check, because don`t exist 8b in figure 1B

Table 1- Make it easier to read the table on compounds column (without having to go back and read) e.g. free anisic acid (1a)

Line 122 – 1na - Why does it only put the number 1in line 1a ?

Line131- No need to define ROS again

Line132- define DCF

Figure 2 A, B  It should make the graph more perceptible (e.g. bar graph 1205 Lu)

Line 157, 161- sub G1 phase, G2/M – define

Figure4B, D, E- Is possible to improve the definition of graphics?

please could you check the pagination13 to 20

Material and Methods

Line 359, 397, 424 - missing supplier

References

standardising the bibliography, for ex 25,26,27

Author Response

1. I think that they should update the bibliography, which is a little outdated, Line 41 -For example regarding the antioxidant properties [1] dates from 1999 and there is recent bibliography about this subject. I propose updated the bibliography.

The bibliography was checked and updated.

2. Line 75-76 “Recently Gliszczynska et al. [20,21]”- These articles are dated 2016-2017, I think that is not recently?

The sentence was modified according to the Reviewer’s suggestion.

3. Line 102 – “and 2a” – check if it is 2a or 1b ?

It should be 1a and 1b – corrected in the text.

4. Line 111- “8a and 8b- figure 1B “ -check, because don`t exist 8b in figure 1B

Corrected.

5. Table 1- Make it easier to read the table on compounds column (without having to go back and read) e.g. free anisic acid (1a)

The table was modified to be easier to follow.

6. Line 122 – 1na - Why does it only put the number 1in line 1a ?

The table was modified to be easier to follow.

7. Line131- No need to define ROS again

Deleted.

8. Line132- define DCF

Defined.

9. Figure 2 A, B It should make the graph more perceptible (e.g. bar graph 1205 Lu)

We agree with the Reviewer that Figure 2 should be more readible. It was modified.

10. Line 157, 161- sub G1 phase, G2/M – define

The definitions of cell cycle phases were added to the text.

11. Figure4B, D, E- Is possible to improve the definition of graphics?

The figures showing the results of cell cycle and apoptosis experiments were thoroughly modified to improve their resolution.

12. please could you check the pagination13 to 20

We see the problem, we hope that the editorial office can help to solve it.

13. Line 359, 397 424 - missing supplier

Missing information added.

14. standardising the bibliography, for ex 25,26,27

The bibliography was checked and standarized.

Reviewer 2 Report

The article " Conjugation with phospholipids as a modification increasing anticancer activity of phenolic acids in metastatic melanoma –  in vitro and in silico studies " presents many results about the conjugation of anticancer drugs with phospholipids being an effective strategy to enhance compounds’ bioavailability in biological systems. This paper is very well written, and it was clear the proposal which the authors intend to show about their findings. Therefore, I recommend publishing after a minor revision by the authors.

Some Comments:
 - The authors should improve this article added new references (less than 30% of references are last five years).
 - Linea 146: Are you sure if the y-axis was named correctly?
 - Linea 174: Some information in Figures 3 and 4 (B, D, and F) is illegible due to lack of image quality. Please, try to do better.
- Linea 211: Figure 5 is unnecessary, in my opinion.
 - Linea 450: In “Statistica 10 software,” add company and country

Author Response

  1. The authors should improve this article added new references (less than 30% of references are last five years).
    The bibliography was checked and updated.
  2.  Linea 146: Are you sure if the y-axis was named correctly?
    The name of Y axis in Figure 2 was changed.
  3. Linea 174: Some information in Figures 3 and 4 (B, D, and F) is illegible due to lack of image quality. Please, try to do better.
    The figures showing the results of cell cycle and apoptosis experiments were thoroughly modified to improve their resolution.
  4. Linea 211: Figure 5 is unnecessary, in my opinion.
    Figure 5 was deleted.
  5. Linea 450: In “Statistica 10 software,” add company and country
    Missing information added.
